# Differentiation Capacity of Porcine Skeletal Muscle-Derived Stem Cells as Intermediate Species between Mice and Humans

**DOI:** 10.3390/ijms24129862

**Published:** 2023-06-07

**Authors:** Tetsuro Tamaki, Toshiharu Natsume, Akira Katoh, Nobuyuki Nakajima, Kosuke Saito, Tsuyoshi Fukuzawa, Masayoshi Otake, Satoko Enya, Akihisa Kangawa, Takeshi Imai, Miyu Tamaki, Yoshiyasu Uchiyama

**Affiliations:** 1Muscle Physiology and Cell Biology Unit, Tokai University School of Medicine, 143 Shimokasuya, Isehara 259-1193, Japan; natsumetoshiharu@gmail.com (T.N.); akiraka@tokai-u.jp (A.K.); nakaji.n@is.icc.u-tokai.ac.jp (N.N.); kousukesaitou427@yahoo.co.jp (K.S.); fukuzawa-tsuyoshi@tokai-u.jp (T.F.); tokai-takeshi@tokai-u.jp (T.I.); 1cmud011@cc.u-tokai.ac.jp (M.T.); y-uchi@is.icc.u-tokai.ac.jp (Y.U.); 2Department of Physiology, Tokai University School of Medicine, 143 Shimokasuya, Isehara 259-1193, Japan; 3Department of Urology, Tokai University School of Medicine, 143 Shimokasuya, Isehara 259-1193, Japan; 4Department of Otolaryngology, Tokai University School of Medicine, 143 Shimokasuya, Isehara 259-1193, Japan; 5Department of Radiation Oncology, Tokai University School of Medicine, 143 Shimokasuya, Isehara 259-1193, Japan; 6Swine and Poultry Research Center, Shizuoka Prefectural Research Institute of Animal Industry, 2780 Nishikata, Kikugawa 439-0037, Japan; micropig@sp-exp.pref.shizuoka.jp (M.O.); satoko1_enya@pref.shizuoka.lg.jp (S.E.); iza.dai2syougai.1koshide@gmail.com (A.K.); 7Department of Orthopedic Surgery, Tokai University School of Medicine, 143 Shimokasuya, Isehara 259-1193, Japan

**Keywords:** micro-mini pig, large animal experiment, GFP-transgenic pig, multipotent stem cells, skeletal muscle, nerve-muscle regeneration

## Abstract

Large animal experiments are important for preclinical studies of regenerative stem cell transplantation therapy. Therefore, we investigated the differentiation capacity of pig skeletal muscle-derived stem cells (Sk-MSCs) as an intermediate model between mice and humans for nerve muscle regenerative therapy. Enzymatically extracted cells were obtained from green-fluorescence transgenic micro-mini pigs (GFP-Tg MMP) and sorted as CD34+/45− (Sk-34) and CD34−/45−/29+ (Sk-DN) fractions. The ability to differentiate into skeletal muscle, peripheral nerve, and vascular cell lineages was examined via in vitro cell culture and in vivo cell transplantation into the damaged tibialis anterior muscle and sciatic nerves of nude mice and rats. Protein and mRNA levels were analyzed using RT-PCR, immunohistochemistry, and immunoelectron microscopy. The myogenic potential, which was tested by Pax7 and MyoD expression and the formation of muscle fibers, was higher in Sk-DN cells than in Sk-34 cells but remained weak in the latter. In contrast, the capacity to differentiate into peripheral nerve and vascular cell lineages was significantly stronger in Sk-34 cells. In particular, Sk-DN cells did not engraft to the damaged nerve, whereas Sk-34 cells showed active engraftment and differentiation into perineurial/endoneurial cells, endothelial cells, and vascular smooth muscle cells, similar to the human case, as previously reported. Therefore, we concluded that Sk-34 and Sk-DN cells in pigs are closer to those in humans than to those in mice.

## 1. Introduction

In regenerative medicine, the scientific basis of therapy is usually obtained from the results of rodent experiments, and is then developed, applied, and pursued in humans. Stem cell transplantation is a common practice in regenerative medicine, and the study of in vivo transplantation is important to examine the cellular capacity for engraftment, differentiation, and contribution to tissue reconstitution with functional recovery. For this purpose, a trace of grafted cells is required. A green fluorescent transgenic (GFP-Tg) mouse has been developed [1] and is widely used to obtain labeled cells. Detailed cellular engraftment, migration, differentiation, and contribution to tissue reconstitution can be established in recipient animals. Subsequently, tissue reconstitution leads to physiological functional recovery, which is the logical basis for human clinical regeneration therapy.

In the next step, the preclinical aspects of regenerative medicine and large-scale animal experiments are as inevitable and important as a transrational animal study for human therapy to confirm the safety and efficacy of cell transplantation. In this case, cellular and tissue regenerative confirmation is difficult and tends to be given less consideration due to difficulties in tracing the transplanted cells. Consequently, we focused on micro-mini pigs (MMP) for their convenient body size [2] and experimental design [3,4,5]. Similarly, GFP-Tg Jinhua pigs were developed as an imaging source for in vivo experiments [6], and GFP-Tg MMP were further developed using the back-crossing method [7].

On the other hand, we have also identified skeletal muscle-derived multipotent stem cells (Sk-MSCs) and sorted them using fluorescence activated cell sorting (FACS) as CD45−/CD34+ (Sk-34) cells [8] and CD45−/CD34− (Sk-DN) cells [9] from a mouse. Using GFP-Tg mice, the in vivo capacity for engrafting and differentiation and their capacity for tissue reconstruction with a good contribution to functional recovery has also been established [10,11,12]. Both mouse Sk-34 and Sk-DN cells exhibited multipotent differentiation capacity to the muscle (skeletal and cardiac), peripheral nerve (Schwann and perineurial/endoneurial), and vascular (vascular smooth muscle and endothelial cells and pericytes) cell lineages, contributing to the three tissue reconstructions with good functional recovery. Importantly, the above cellular multipotency was observed in both mouse Sk-34 and Sk-DN cells, although Sk-DN cells are considered hierarchically upstream of Sk-34 cells [13].

Using the same cell fractionation technique, the above multiple cell differentiation and tissue reconstruction capacities with functional recovery were also observed in human Sk-34 and Sk-DN cells [14]. However, human Sk-DN cells show a specific capacity for the skeletal muscle cell lineage, whereas Sk-34 cells show specific peripheral nerve and vascular cell lineage differentiation [14]. This difference between mice and humans is interesting and should clarify the importance of intermediate animals, such as MMPs [3,4,5], in stem cell biology, physiology, and other preclinical studies of large animals.

## 2. Results

### 2.1. Fractionation of Sk-34 and Sk-DN Cells

The sorting patterns of the porcine Sk-34 and Sk-DN cells are shown in Figure 1. CD45+ cells were first excluded as hematopoietic cells, and CD45− cells were further sorted by CD34 and CD29 (Figure 1). The Sk-DN cells were re-established as CD45−/CD34−/CD29+ cells for their possible positive selection and debris removal. The results showed that Sk-34 and Sk-DN cells exist in the pig skeletal muscle, as observed in previous studies on mice and humans.

### 2.2. Expression of Skeletal Muscle, Peripheral Nerve, and Vascular Cell Lineage Specific mRNAs Immediately after Sorting

The expression of specific mRNA in Sk-34 and Sk-DN cells was measured immediately following cell sorting to detect the in vivo status of the cells (Figure 2). Green color, which specifically represents myoblasts and skeletal muscle relative factors, is dominant in Sk-DN cells. Lower expression of Pax7, MyoD, M-Cadherin, and CACNB1 (voltage-gated L-type calcium channel beta-1) was observed in Sk-34 cells. However, no differences were observed in the markers of peripheral nerves and vascular cells. Non-expression of CD34 in Sk-DN cells was confirmed at the mRNA level.

### 2.3. Distributions of Pax7 and MyoD Positive Cells after Culture

The expression of Pax7 and MyoD was also examined at the protein level after 5–6 days of cell culture. The percentages of Pax7+ and MyoD+ cells among Sk-34 and Sk-DN cells are shown in Figure 3. At the protein level, the expression of Pax7/MyoD was significantly higher (approximately three-fold) in Sk-DN cells, consistent with the mRNA results. Therefore, the ability of Sk-DN cells to differentiate into the skeletal muscle cell lineage was stronger.

### 2.4. In Vivo Differentiation Capacity of Sk-DN Cells

The in vivo differentiation capacity of Sk-DN cells was confirmed after cell transplantation into damaged TA muscle and sciatic nerve models of nude mice and rats. Firstly, there were no Sk-DN cells engrafted in the damaged sciatic nerve, as one of their characteristics was shown in human Sk-DN cells. However, a large number of GFP+ myofibers (Sk-actin-positive) with large and small diameters were detected in the damaged TA muscles of mice (Figure 4A). These GFP+ fibers had central nuclei, which are characteristic of regenerative fibers (Figure 4B–D). Laminin production was detected on fibers with small diameters, indicating the presence of newly formed fibers (Figure 4C,D). Similarly, GFP+ reactions were observed inside the parent fiber, indicating the possible involvement of satellite cells (Figure 4C, oblong line).

### 2.5. In Vivo Differentiation Capacity of Sk-34 Cells

The in vivo differentiation capacity of Sk-34 cells was confirmed after cell transplantation into the damaged TA muscle and sciatic nerve (Figure 5). GFP+ skeletal muscle fibers (GFP+/Sk-actin+) were also detected in the mouse TA damage model (Figure 5A).

Several GFP+/p75+ cells were evident in the rat sciatic nerve injury model (Figure 5B, yellow arrows), indicating the presence of pig Sk-34 cell-derived Schwann cells. Similarly, the GFP+ perineurium/endoneurium were clearly observed in the immunoelectron micrographs (Figure 5C,D). Myelinated/nonmyelinated axons were frequently encircled by GFP+ dark reactions. The encirclements could be detected on single axons as an endoneurium with a clear nucleus (endoneurial cells, Figure 5D) and on multiple axons as a perineurium (perineurial cells, Figure 5C). The same reaction was also evident around the capillary endothelial cells, possibly apart from the pericytes (Figure 5C).

Vascular cells, GFP+/CD31+ endothelial cells (Figure 5E, arrows), and GFP+/αSMA+ vascular smooth muscle cells (Figure 5F, arrows) were observed in the mouse TA injury model. Consequently, porcine Sk-34 cells exhibited the multipotent differentiation capacity of skeletal muscle, peripheral nerve, and vascular cell lineage cells.

In the in vivo transplantation experiment with both Sk-DN and Sk-34 cells, GFP-positive reactions were not detected in the rat TA injury model. A few positive reactions were observed after the enhancement of GFP by the anti-GFP antibody, but the reactions were quite weak in contrast to those in the other models. Therefore, we assumed that the GFP-Tg MMP emission intensity was lower than that of a GFP-Tg mouse and that these results could depend on the size of the recipient tissue.

## 3. Discussion

The present study clearly indicated that porcine Sk-DN cells were highly myogenic cells that were able to form skeletal muscle fibers after transplantation of damaged TA muscle. There were probably primary myoblasts, which were able to form new fibers with the basal lamina (laminin+) and satellite cells. However, they are not capable of regenerating peripheral nerves and blood vessels because of the lack of cell differentiation capacities in their related cell lineages, and this similar to the previous human case. In contrast, porcine Sk-34 cells exhibited a multipotent differentiation capacity of the skeletal muscle (primary myoblasts and satellite cells), peripheral nerve (Schwann cells, perineurial/endoneurial cells), and vascular (endothelial cells, vascular smooth muscle cells, and probably pericytes) cell lineages. These characteristics are similar to those observed in a previous mouse case.

Previously, we identified multipotent Sk-34 and Sk-DN cells in mouse skeletal muscles [8,9]. Using GFP-Tg mice and cell transplantation, the in vivo capacity for engrafting, differentiation, and tissue reconstruction with a good contribution to functional recovery has been established [10,11,12]. In particular, both Sk-34 and Sk-DN cells exert multipotent differentiation capacity in the muscle (skeletal and cardiac), peripheral nerve (Schwann and perineurial/endoneurial cells), and vascular (vascular smooth muscle and endothelial cells, and pericytes) cell lineages, contributing to the three kinds of tissue reconstructions with good functional recoveries [10,11,12,15,16]. Importantly, the above cellular multipotency was equally observed in both Sk-34 and Sk-DN cells in mice, although Sk-DN cells are considered hierarchically upstream of Sk-34 cells [13].

Thereafter, we confirmed that Sk-34 and Sk-DN cells were also present in the human skeletal muscle, which could be obtained using the same method of enzymatic extraction and FACS sorting as the mouse. However, in humans, these two cell fractions show different in vitro and in vivo characteristics [14]. In other words, the same cell fractions of Sk-34 and Sk-DN exist in humans, but the two cell fractions have different differentiation capacities. Sk-DN cells are specific for the skeletal muscle cell lineage, whereas Sk-34 cells are multipotent lineages of peripheral nerves and vascular cells. Therefore, the differences between humans and mice should be properly clarified for the therapeutic purposes of regenerative medicine.

This difference can be assumed to simply depend on the evolutionary change of the biological species. However, in this type of cross-species comparison study, the coherence of the cellular fraction method is important, and this is the first objective of this study. The cell fractionation method used in the present pig study strictly followed that used in previous mouse and human studies. Thus, it is possible that the results reflect the differences between species. Pig Sk-34 and Sk-DN cells are similar to human cells because of the dominant myogenic differentiation of Sk-DN cells in vivo, which is supported by the in vitro cell culture results (Figure 3) and mRNA analysis immediately after cell extraction and sorting (Figure 2). Interestingly, Sk-DN cells did not engraft into the damaged sciatic nerve (peripheral nerve circumference). This is the same as in humans and different from mice; this is the second point of the present study. A slightly different aspect is the myogenic potential of Sk-34 cells (Figure 5). In humans, myogenic differentiation was not detected in Sk-34 cells [14], whereas little myofiber formation was observed in pig Sk-34 cells transplanted in vivo (Figure 5). This trend was also supported by the results of the in vitro cell culture (Figure 3).

Another issue that must be resolved is whether Sk-DN cells include myogenic cells other than satellite cells. In our series of previous studies, satellite cells were CD34 negative; thus, they were included as Sk-DN cells. Therefore, the next question was whether satellite cells are capable of new myofiber formation. Skeletal muscle fibers (myofibers) are necessary for the formation of the basal lamina. However, satellite cells localize within the basal lamina of parent myofibers to regenerate/generate cytoplasmic myofibrils. Therefore, the notion that satellite cells do not necessarily form or produce the basal lamina is correct. However, interstitial myogenic cells must form the basal lamina when new fibers are formed (fiber hyperplasia) in the interstitial spaces. In fact, small fibers with basal lamina (laminin+) were also detected after Sk-DN cell transplantation (Figure 4B–D). In addition, GFP+ cells were detected inside the basal lamina of parent fibers (Figure 4C). In the above regard, it is possible to consider that Sk-DN cells include primary myoblasts and satellite cells. Possible satellite cell-like reactions were not observed during the transplantation of Sk-34 cells, which showed slight myogenic potential.

However, the regenerative capacity of the peripheral nerve of Sk-34 cells was quite similar to that of humans (Figure 5) and not very different from that of mice [17]. Active regeneration of the perineurium/endoneurium with differentiation into perineurial/endoneurial cells was also observed (Figure 5B–D). Therefore, it may also have therapeutic potential for long-gap peripheral nerve injuries, as shown in both humans and mice [17,18,19].

Stem cells derived from skeletal muscles, other than satellite cells, have been identified and fractioned using various methods and studied for their therapeutic potential over the last 25 years [20,21,22,23,24,25,26,27,28,29,30]. This is because skeletal muscle usually accounts for 40–60% of the lean body mass in humans and is situated in the outer part of the body next to the skin and subcutaneous fat tissues. Therefore, tissue sampling to obtain an autologous stem cell source is considered relatively easy and safe from a therapeutic perspective.

Basic studies on stem cells have been initiated and developed mainly using mouse skeletal muscles, and the results have been applied to humans on theoretical grounds. It appears that the in vivo cell behavior and differentiation capacity of skeletal muscle-derived stem cells (Sk-MDSCs) are somewhat different between mice and humans after cell transplantation. Given this difference, it can be assumed that the differences in the cell fractionation method in each study were largely affected. This is because there are variations in cell fractionation methods, and several differences in cell differentiation abilities have been observed among mouse studies [25,31,32,33,34,35,36,37,38]. Therefore, the present study accomplished methodological consistency, including cell isolation, fractionation, and expansion, and analyzed cell differentiation capacity using the same in vivo transplantation model. Consequently, it is considered that the present results show the transitional phase of evolutionary change between mice and humans. Whether this change is positive or negative is unclear. However, given the greater ability of the mouse skeletal muscle to regenerate, it is likely negative. Therefore, intermediate large-animal experiments are considered a valuable step in stem cell biology and physiology. Transplantation of MMP-to-MMP Sk-34 cells for nerve regeneration is of great interest, especially in the therapeutic use of preclinical experiments on large animals. However, low green fluorescent protein (GFP) emissions within GFP-Tg MMP are a concern.

## 4. Materials and Methods

### 4.1. Animal Usage

In the present study, we used micro-mini pigs (MMP) produced by the Swine and Poultry Research Center, Shizuoka Prefectural Research Institute of Animal Industry. Green fluorescent protein transgenic (GFP-Tg) MMP was used as the donor animal in the in vivo transplantation experiment, and non-Tg MMP was used for in vitro analysis. This GFP-Tg MMP was prepared via the backcross method [7] based on GFP-Tg Jinhua pigs [6] and MMP [3,4,5].

GFP-Tg MMP (12–14-week-old, *n* = 4) and/or non-Tg MMP (12–14-week-old, *n* = 4) were anesthetized with an overdose of pentobarbital (100–120 mg/kg, IP), and deep sleep was induced. After blood removal, muscle samples (80–100 g) were collected from the vastus lateralis. All the study protocols were approved by the Tokai University School of Medicine Committee on Animal Care and Use (#210421).

### 4.2. Cell Isolation and Sorting

Sk-34 and Sk-DN cells were isolated using the same method as previously described for mouse and human muscles [8,9,14,18], and methodological consistency was achieved across three species. Briefly, muscle samples were washed several times with Dulbecco’s modified essential medium (DMEM) supplemented with 1% penicillin/streptomycin and cut into several pieces (5–7 mm in thickness and width and 30–40 mm in length). It should be noted that the muscles were never minced. The muscle pieces were washed again in DMEM, then treated with 0.1% collagenase type IA (Sigma-Aldrich, St. Louis, MO, USA) in DMEM containing 7.5% fetal calf serum (FCS) with gentle agitation for 2 h at 37 °C. Extracted cells were filtered through 70-µm, 40-µm, and 20-µm nylon strainers to remove muscle fibers and other debris. Then, cells were frozen at −80 °C using a biofreezing vessel (BICELL; Nihon Freezer Co., Ltd., Tokyo, Japan) and stored in liquid nitrogen using cell preservative solution (Cell Banker; Juji-field, Tokyo, Japan) until use. This process is identical to that in human cells [14].

### 4.3. Cell Sorting

For cell sorting, stored cells were thawed and resuspended in Iscove’s modified Dulbecco’s medium (IMDM) containing 10% FCS and prepared for cell sorting. Suspended cells were stained with CD34 (goat anti-pig CD34 polyclonal, R&D, Minneapolis, MN, USA) and (Rabbit anti-goat Alexa 594, Molecular Probes, Eugene, OR, USA), CD45 (Mouse anti-pig monoclonal, Bio-Rad, Tokyo, Japan), (rabbit anti-mouse, Alexa 647), and CD29 (mouse anti-pig CD29 Alexa 647 conjugated, BD Bioscience, San Jose, CA, USA) antibodies, and Sk-34 (CD45−/34+) and Sk-DN (CD45−/34−/29+) were obtained using FACS Aria III (Nippon BD, Tokyo, Japan).

### 4.4. RT-PCR

To test the putative cell differentiation capacity, the expression of specific markers for skeletal muscle, peripheral nerve, and vascular cell lineages, as well as neurotrophic and vasculogenic factors were examined with RT-PCR for Sk-34 and Sk-DN/29+ cells immediately after cell sorting. The specific primers for pig cells and the analyzed materials are summarized in Appendix A. Cultured cells were expanded for 5–6 days, strictly corresponding to the method previously reported for human cells [14,18], lysed, and total RNA was purified using a QIAGEN RNeasy Micro Kit (Molecular Probes, Eugene, OR, USA). First-strand cDNA synthesis (QIAGEN, Hilden, Germany) was performed with an Invitrogen SuperScript III system using a dT30-containing primer (see above), and specific PCR (35 cycles of 30 s at 94 °C, 30 s at 60–65 °C, and 2 min at 72 °C) was performed in a 15-µL volume containing Ex-Taq buffer, 0.8 U of ExTaq-HS-polymerase, 0.7 µM specific sense and antisense primers, 0.2 mM dNTPs, and 0.5 µL of cDNA. To determine a slight difference in the expression intensity on the support, additional specific PCR cycles of 30 and 40 were performed (30, 35, and 40 cycles).

The analysis was repeated three times, and the relative expression intensity of each band was qualitatively classified into three levels: an apparently strong (+3), an apparently low (+1), an intermediate (+2), or an undetectable (0) band with respect to the housekeeping control gene (GAPDH), which was used for each electrophoresis.

### 4.5. In Vitro Myogenic Differentiation Capacity of Pig Sk-34 and Sk-DN Cells

First, the in vitro myogenic differentiation abilities of Sk-DN and Sk-34 cells were determined via immunocytochemistry using a cytospin preparation. This analysis was performed using non-transgenic MMP (*n* = 4). Both cell fractions were cultured following a previously reported method for human-specific conditions [14], and cellular behaviors were observed.

After 5–7 days of culture, both cell types were removed using trypsin-EDTA, washed with DMEM/20% FCS, and suspended in DMEM. Suspended cells were fixed with 2% paraformaldehyde (PFA)/0.1 M phosphate buffer (PB) for 10 min, spun, applied to a glass slide (cytospin preparation), re-fixed with 4% PFA/PB for 10 min, and washed with 0.1 M phosphate-buffered saline (PBS). The samples were then treated with 10–20% sucrose/PBS and frozen. After several freeze–thaw cycles, the samples were prepared for immunostaining to analyze myogenic differentiation. Monoclonal anti-MyoD (1:50, 4 °C overnight; Dako, Carpinteria, CA, USA) antibodies were used to detect myogenic cells; monoclonal anti-Pax7 antibodies (1:50, 4 °C overnight; Developmental Studies Hybridoma Bank, University of Iowa, Iowa, IA, USA) were used to detect putative satellite cells.

The proteins were visualized using Alexa Fluor-594-conjugated goat anti-mouse antibodies (Molecular Probes, Eugene, OR, USA). Quantitative analysis was performed using the Stereo Investigator software (MBF Bioscience, Williston, VT, USA, https://mbfbioscience.jp/stereoinvestigator.html, 30 January 2011) and Photoshop 2023 (Adobe Systems Inc., San Jose, CA, USA). Data were averaged and expressed as percentages (positive cells/total cells), and the cells that included more myogenic cells were compared.

### 4.6. In Vivo Cell Differentiation Capacity and Recipient Animals

To determine the in vivo differentiation potential of Sk-DN and Sk-34 cells, athymic nude mice (female, BALB/cA Jcl-*nu*/*nu*; CLEA Japan, Tokyo, Japan, age 5–6 wk, *n* = 22) and rats (male and female, F344/NJcl-mu/mu; CLEA, Tokyo, Japan, age 8–12 wk, *n* = 14) were used as recipient animals. The donor cells were obtained from GFP-Tg MMP (*n* = 4), and the cells from the 4 muscle samples were transplanted into the two types of models of rats and mice described below. All experimental procedures were approved by the Tokai University School of Medicine Committee on Animal Care and Use (#210421, #214022, and #215010).

We then used 2 types of animal models: (1) the “severe muscle damage model for tibialis anterior (TA)”, which was largely removed muscle tissue with nerve and blood vessel branches [10], and (2) the “severe crash injury model for sciatic nerve” as a simulation of Seddon’s axonotmesis and/or Sunderland’s fourth degree nerve damage, which involves loss of axons, endoneurial tubes, perineurial fasciculi, and vascular networks, while continuity of the epineurium is maintained [17]. Therefore, multiple stimulations of skeletal muscle-related cells (such as satellite cells and other myogenic cells), peripheral nerves (Schwann cells and endoneurial/perineurial cells), and vascular lineage cells (pericytes, endothelial cells, and smooth muscle cells) can be expected.

Briefly, in the severe TA muscle injury model, the fascia of the left TA muscle of nude mice and/or rats was exposed by skin incision, and the fascia was minimally cut. Using forceps, muscle fibers with nerves and blood vessels from the region surrounding the motor point of the TA muscle were manually removed to 30–40% of the whole muscle mass. To avoid diffusion of transplanted cells, the fascia incision was sutured and then Sk-DN or Sk-34 cells (about 5 × 10^5^ in mice, 1 × 10^6^ in rats) suspended in 2~4 µL of DMEM, were slowly injected into the damaged muscle portion using a fine-tip glass micropipette. The skin was then sutured, and a transparent sterile/analgesic plastic dressing (Nobecutan spray; Yoshitomi Chemical, Tokyo, Japan) was sprayed over the wound.

A nerve crush injury model was established using nude mice and rats. The right sciatic nerve was exposed through skin and gluteal muscle incisions. The sciatic nerve was then repeatedly crushed to 7 mm in mice and 12 mm in rats along the longitudinal axis using forceps. In this case, most of the peripheral nerve support tissues were destroyed, except for the epineurium, which is the outermost layer of the nerve. Cells were suspended in DMEM at a concentration of 2.5 × 10^5^ cells/2 μL, and 5 × 10^5^ cells/3 μL and injected into the destroyed hollow portion of the nerve through the remaining epineurium using a fine-tip glass pipette [17,18].

### 4.7. Immunohistochemistry and Immunoelectron Microscopy

In vivo cell differentiation potential was examined using immunohistochemistry and immunoelectron microscopy. After 5 weeks of transplantation, the recipient animals were anesthetized with pentobarbital (60 mg/kg) and perfused with warm PBS through cannulation from the abdominal aorta to remove circulating blood, followed by perfusion fixation with 4% PFA/PB. After fixation, the TA muscles or sciatic nerves were removed and re-fixed with 4% PFA/PB overnight, followed by washing with PBS and treatment with a graded 5–25% sucrose/PBS series. The TA muscles were quickly frozen using isopentane pre-cooled with liquid nitrogen, and the nerves were embedded in OCT compound and frozen at −80 °C. Both samples were stored at −80 °C until use.

For staining of the histological sections, several 7 µm cross-sections of the operated TA muscles and sciatic nerves were obtained. Engrafted GFP-Tg pig cells were enhanced using a rabbit anti-GFP IgG fraction (1:300, A11122; Molecular Probes, Inc., Eugene, OR, USA).

Blood vessels were detected using mouse monoclonal α-smooth muscle actin (SMA, Cy3-conjugated directly; 1:1500, for 1 h at room temperature; Sigma, St. Louis, MO, USA), rat anti-mouse CD31 (1:500, 4 °C overnight; BD Biosciences, San Jose, CA, USA), and rat monoclonal anti-pig CD31/PECEM-1 (1:200, for 2 h at room temperature; R&D, Minneapolis, MN, USA). Immature and/or newly differentiated Schwann cells were detected using rabbit anti-p75 polyclonal antibody (1:400, 4 °C overnight; CST, Boston, MA, USA). Muscle fibers were confirmed using rabbit polyclonal anti-skeletal muscle actin (1:300, for 1 h at room temperature; Abcam, Cambridge, UK). Rabbit polyclonal anti-laminin (Pan-laminin, 1:1500 for 2 h at room temperature, LSL, Tokyo, Japan) and/or rat monoclonal anti-mouse laminin (β-2 chain, 1:2000 for 2 h at room temperature, Chemicon, Temecula, CA, USA) antibodies were used to distinguish between the inside and outside (interstitial spaces) of the muscle fibers. The reactions were visualized using Alexa Fluor-488- and 594-conjugated goat anti-rabbit and anti-rat antibodies (1:500 for 2 h at room temperature; Molecular Probes, Eugene, OR, USA). Nuclei were counterstained with DAPI (4′,6-diamino-2-phenylindole).

The engrafted cells were further analyzed via immunoelectron microscopy. Cryosections were stained using anti-GFP (1:50, 4 °C overnight; clone 235-1, biotin conjugate; Millipore, Burlington, MA, USA), followed by HRP-conjugated streptavidin secondary antibody (1:200, for 1 h at room temperature; Dako, Carpinteria, CA, USA) to label engrafted pig cells. Reactions were visualized with DAB (3,3’-Diaminobenzidine) after fixation in 1% glutaraldehyde/0.1 M PB. Visualized sections were then fixed in 1% osmium tetroxide/0.05 M PB and prepared for electron microscopic analysis. Cell differentiation into Schwann cells, perineurial/endoneurial cells, vascular endothelial cells, pericytes, and fibroblasts were also observed. A detailed immunoelectron microscopy method has been reported previously [10,11,39].

### 4.8. Statistics

Differences in cytospin data between Sk-DN and Sk-34 cells was analyzed using the parametric Tukey–Kramer post hoc test, and the significance level was set at *p* < 0.05. Values are expressed as mean ± SE.

## Figures and Tables

**Figure 1 ijms-24-09862-f001:**
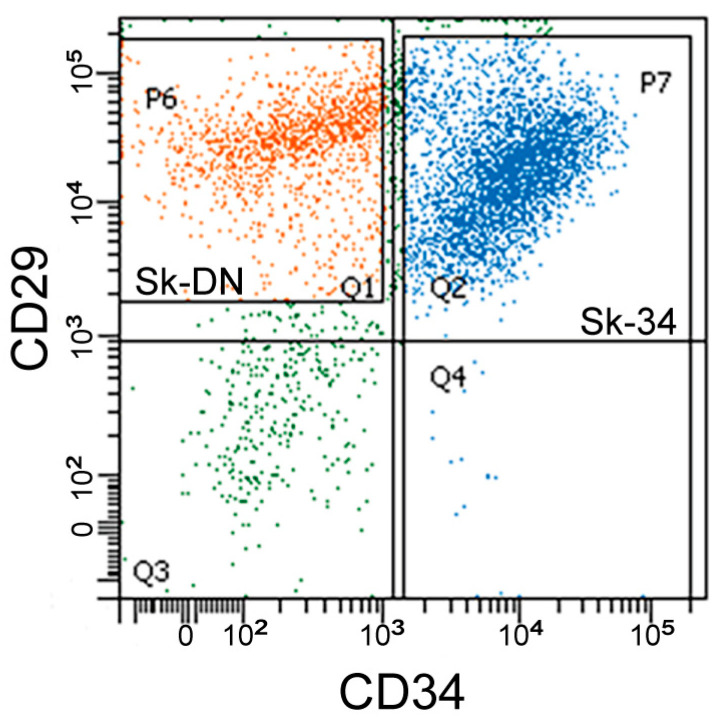
Sorting pattern of pig Sk-34 and Sk-DN cells. Sk-DN cells = P6 gate (Q1, orange dots), Sk-34 cells = P7 gate (Q2 and Q4, blue dots). Green dots in Q3 gate and others were excluded fraction mainly composed of debris and the other type of cells.

**Figure 2 ijms-24-09862-f002:**
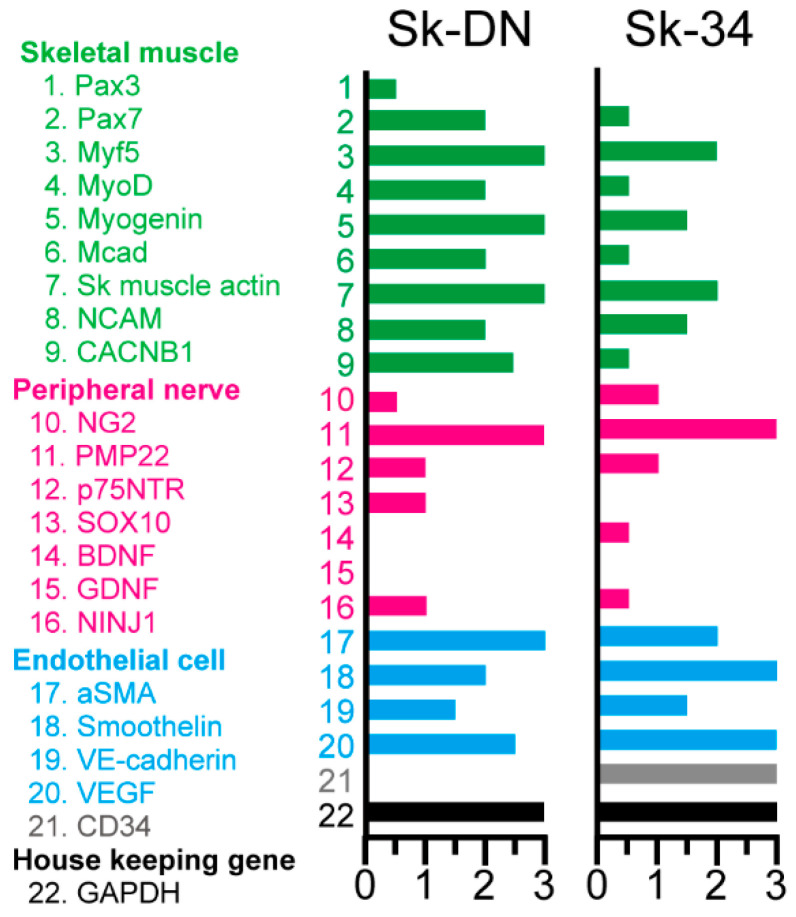
Expression of mRNAs specifically for the skeletal muscle (green; 1~9), peripheral nerve (pink; 10~16) and vascular (blue; 17~20) cell lineages of the Sk-DN and Sk-34 cells. Grey = CD34 (21). Black = GAPDH (housekeeping gene; 22). Expression was evaluated between 0 to 3 based on the value of GAPDH, and 3 is the highest. Samples were obtained from the non-GFP-Tg-MMP (*n* = 4). In this case, the total RNA extracts were preliminary mixed (averaged) and analyzed.

**Figure 3 ijms-24-09862-f003:**
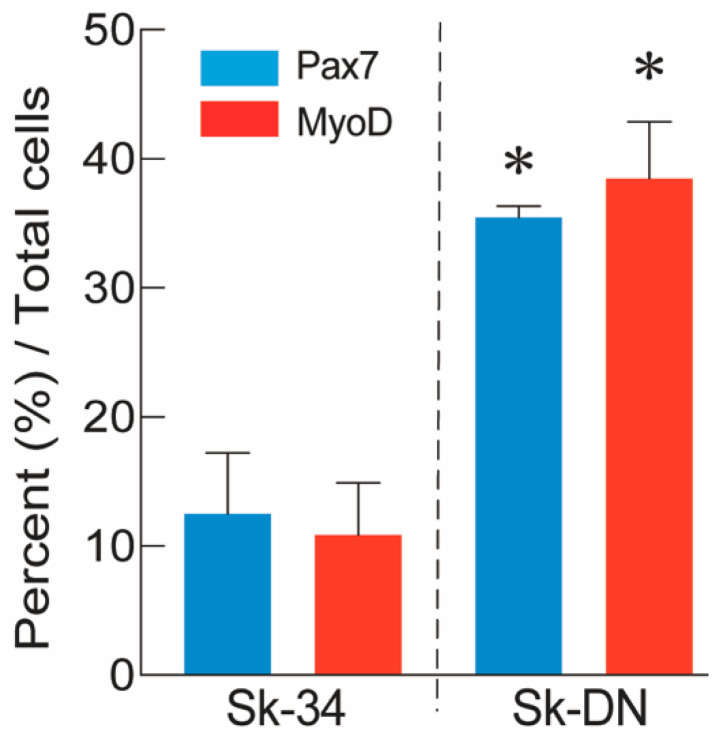
Percent distribution of Px7+ and MyoD+ cells in the Sk-34 and Sk-DN cells based on total cells. The non-GFP-Tg-MMP was used in this analysis (*n* = 4), and 2–3 cytospin preparations/animal were counted. Values are expressed mean ± S.E. * *p* < 0.05.

**Figure 4 ijms-24-09862-f004:**
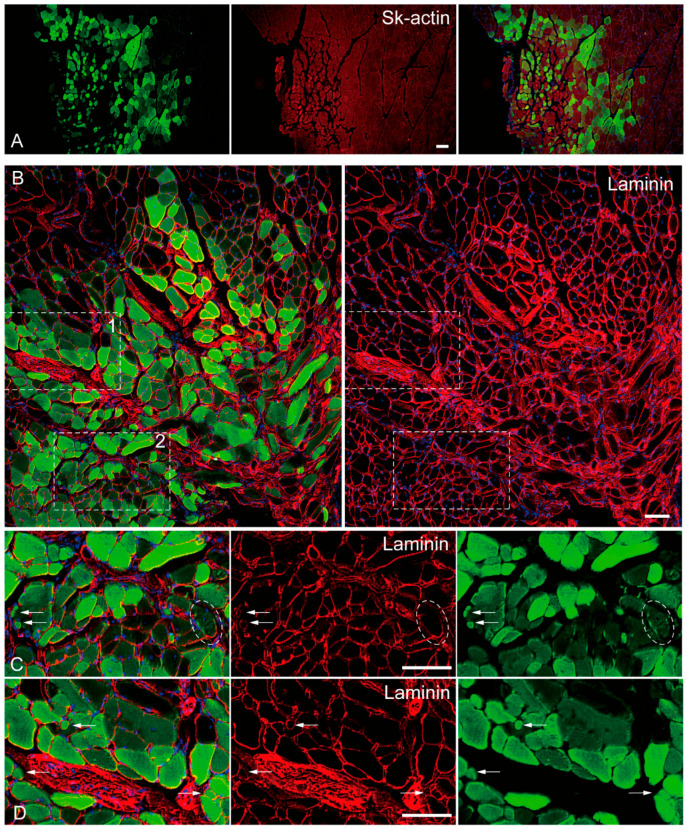
In vivo differentiation capacity of Sk-DN cells. Samples were obtained from the mouse TA injury model 5-weeks after operation. (**A**): anti-Skeletal muscle actin (red staining), (**B**–**D**): anti-Laminin staining (red staining). Inset 1 and 2 (dotted squares) in panel (**B**) correspond to the part as panel (**C**,**D**). Arrows in (**C**,**D**) shows typical small fiber having laminin. Dotted oblong line point to the possible involvement of green cells as satellite cells. Blue staining = DAPI. Bars = 100 μm.

**Figure 5 ijms-24-09862-f005:**
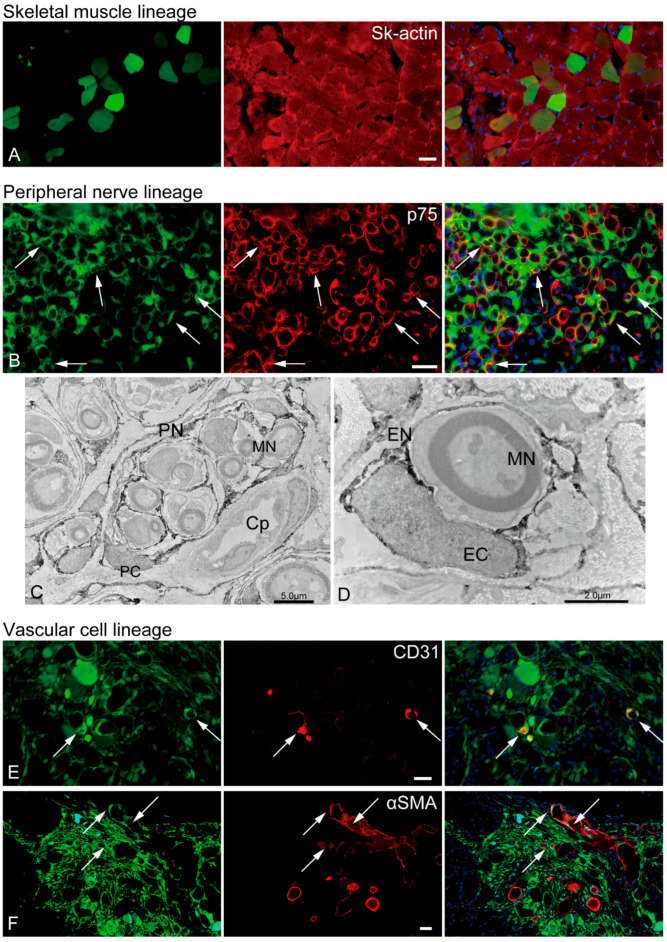
In vivo differentiation capacity of Sk-34 cells. The results of skeletal muscle and vascular lineage are from the mouse TA damage model, and peripheral nerve lineage are from the rat sciatic nerve damage model. In the immunohistochemical photographs (**A**,**B**,**E**,**F**), left column = GFP + cells/tissues; center = reactions for each antibody; right column = merge image. Arrows show typical portions of double reactions with GFP and red reactions for each antibody (yellow colors). Blue staining = DAPI. (**C**,**D**): Immunoelectron microscopy. Dark portions are DAB+ reactions for anti-GFP. PN = perineurium, MN = myelinated nerve, PC = perineurial cell, Cp = capillary, EC = endoneurial cell, EN = endoneurium. Bars = 30 μm.

## Data Availability

Data are contained within the article.

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
