# Peer review of "Differentiation Capacity of Porcine Skeletal Muscle-Derived Stem Cells as Intermediate Species between Mice and Humans"

_ijms, 2023, doi:10.3390/ijms24129862_

Round 1

Reviewer 1 Report

In the manuscript titled “Differentiation capacity of porcine skeletal muscle-derived stem cells as intermediate species between mice and humans” Tamaki and colleagues investigated the ability of Sk-34 and Sk-DN cells to differentiate into skeletal muscle, peripheral nerve and vascular cell lineages in pig, mouse and human experimental model. Both Sk-34 and Sk-DN cells, derived from mouse skeletal muscles, exert multipotent differentiation capacity in the muscle, peripheral nerve and vascular cell lineages. The same cell fraction in pig have different differentiation capacities. The formation of muscle fibers is higher in Sk-DN cells than in Sk-34 cells. In contrast, the capacity to differentiate into peripheral nerve and vascular cell lineages is stronger in Sk-34 cells. The same differentiation capacity was observed in human Sk-34 and Sk-DN cells, concluding that pig skeletal muscle-derived stem cells (Sk-MSCs) as intermediate model between mice and human for nerve regenerative therapy.

Overall, the manuscript is well written.

 The reference list is old and I suggest its updating. 

I think this article is acceptable for publication in International Journal of Molecular Sciences.

Author Response

Response to the reviewer #1

In the manuscript titled “Differentiation capacity of porcine skeletal muscle-derived stem cells as intermediate species between mice and humans” Tamaki and colleagues investigated the ability of Sk-34 and Sk-DN cells to differentiate into skeletal muscle, peripheral nerve and vascular cell lineages in pig, mouse and human experimental model. Both Sk-34 and Sk-DN cells, derived from mouse skeletal muscles, exert multipotent differentiation capacity in the muscle, peripheral nerve and vascular cell lineages. The same cell fraction in pig have different differentiation capacities. The formation of muscle fibers is higher in Sk-DN cells than in Sk-34 cells. In contrast, the capacity to differentiate into peripheral nerve and vascular cell lineages is stronger in Sk-34 cells. The same differentiation capacity was observed in human Sk-34 and Sk-DN cells, concluding that pig skeletal muscle-derived stem cells (Sk-MSCs) as intermediate model between mice and human for nerve regenerative therapy.

Overall, the manuscript is well written.

 The reference list is old and I suggest its updating. 

I think this article is acceptable for publication in International Journal of Molecular Sciences.

Response to the reviewer 1.

We thank the reviewer for the overall positive comments. 

The reviewer is right, and we have to admit that our reference list is basically old. However, this has two important reasons; 1) the original basic of skeletal muscle-derived stem cells, which including identification, isolation and fractionation, have been reported early 2000; 2) a lot of works since then are basically using modified methods, and not a little paper makes no account them. In this regard, we certainly think that we have to pay respect to those of original/first works (reports). Anyway, the list is old is the fact. Therefore, we added citation relatively recent reports similar to the original works together as suggested by the reviewer for the updating.  

Reviewer 2 Report

In this original research report, Tamaki and co-workers describe the isolation and characterization of skeletal-muscle multipotent stem cells (Sk-MSCs) from GFP-transgenic micro-mini pigs, sorted into two CD34+/45- (Sk-34) and CD34-/45-/29+ (Sk-DN) fractions. Since in previous research the authors have found differences in the differentiation capacities of these cells between human and mouse species, the purpose of the present study was to investigate the features of these cells in a large animal model closer to humans. The authors found that, similar to human Sk-MSCs, the porcine Sk-DN cells were restricted to the myogenic lineage and while successfully differentiating into skeletal muscle in vitro as well as in vivo, they failed to generate peripheral nerve and vascular endothelial cells. In contrast, the Sk-34 cells were able to generate all of the three cell lineages. The difference with the mouse model is that mouse Sk-DN counterparts were able to generate all of the skeletal, neural, and vascular cell lineages. The conclusion of the study was that the pig is a more relevant model for human conditions than the rodents, which makes the study significant in terms of the validation of the porcine model.

 On the other side, some important details need to be included in the manuscript before publication:

 -          How many biological replicates (cell fractions from different animals) were isolated and analyzed? In the Methods section, it was stated that 4 transgenic and 4 non-transgenic animals were used, but it is not clear how many biological replicates were used in each analysis. This is not clear in cases of figures 2 and 3, where these numbers need to be shown as (n=…) in the figure legends. The only information obvious from the Methods is that every RT-PCR analysis was performed thrice (technical replicates), but was this only on one or more different fractions?

 -          Similarly, it should be shown in the Methods how many different biological samples were used for transplantation.

 These numbers are important as they show how sound the study data is from a statistical standpoint. Besides that, this reviewer has no further comments.

Overall, the manuscript is clearly written and easy to follow. A few spelling errors (e.g. “(Figdure 5A)” on line 136) were noticed, so an additional spell check is recommended

Author Response

Response to the reviewer #2

In this original research report, Tamaki and co-workers describe the isolation and characterization of skeletal-muscle multipotent stem cells (Sk-MSCs) from GFP-transgenic micro-mini pigs, sorted into two CD34+/45- (Sk-34) and CD34-/45-/29+ (Sk-DN) fractions. Since in previous research the authors have found differences in the differentiation capacities of these cells between human and mouse species, the purpose of the present study was to investigate the features of these cells in a large animal model closer to humans. The authors found that, similar to human Sk-MSCs, the porcine Sk-DN cells were restricted to the myogenic lineage and while successfully differentiating into skeletal muscle in vitro as well as in vivo, they failed to generate peripheral nerve and vascular endothelial cells. In contrast, the Sk-34 cells were able to generate all of the three cell lineages. The difference with the mouse model is that mouse Sk-DN counterparts were able to generate all of the skeletal, neural, and vascular cell lineages. The conclusion of the study was that the pig is a more relevant model for human conditions than the rodents, which makes the study significant in terms of the validation of the porcine model.

 On the other side, some important details need to be included in the manuscript before publication:

  • How many biological replicates (cell fractions from different animals) were isolated and analyzed? In the Methods section, it was stated that 4 transgenic and 4 non-transgenic animals were used, but it is not clear how many biological replicates were used in each analysis. This is not clear in cases of figures 2 and 3, where these numbers need to be shown as (n=…) in the figure legends. The only information obvious from the Methods is that every RT-PCR analysis was performed thrice (technical replicates), but was this only on one or more different fractions?

Response

We thank the reviewer for the very important suggestion, and we totally agree with the reviewer. We added further information in the legends of Figure 2 and 3 (using red letters for easy identification).  

  • Similarly, it should be shown in the Methods how many different biological samples were used for transplantation. These numbers are important as they show how sound the study data is from a statistical standpoint.

Response

We added following phrase in the Methods as suggested by the reviewer.

“The donor cells were obtained from GFP-Tg MMP (n=4), and the cells from the 4 muscle samples were transplanted into the two types of models of rats and mice described below. “

  • Besides that, this reviewer has no further comments.

Response

Again, we thank the reviewer for overall positive comments. 

Comments on the Quality of English Language

Overall, the manuscript is clearly written and easy to follow. A few spelling errors (e.g. “(Figdure 5A)” on line 136) were noticed, so an additional spell check is recommended

Response

We thank the reviewer for the kind checking of our manuscript. We have done careful spell checking again.